# L467F;F508del Complex Allele in a Heterozygous State with CFTRdele2,3: What to Expect from CFTR Modulators?

**DOI:** 10.3390/ijms262311742

**Published:** 2025-12-04

**Authors:** Elena Kondratyeva, Anna Efremova, Yuliya Melyanovskaya, Maria Krasnova, Michael Milovanov, Olga Shchagina, Anna Stepanova, Anna Voronkova, Victoria Sherman, Sergey Kutsev, Dmitry Goldshtein

**Affiliations:** Research Centre for Medical Genetics, 1 Moskvorechye St., 115552 Moscow, Russia

**Keywords:** cystic fibrosis, *CFTR* gene, targeted therapy, pancreatitis, intestinal organoids, intestinal current measurement, L467F, F508del, complex allele

## Abstract

CFTR modulators have significantly affected the prognosis for cystic fibrosis, improving the clinical course in most patients with the F508del variant and several other *CFTR* gene variants. The presence of complex alleles, including more than one variant in the cis position, can change the properties of the protein and the efficacy of modulators. Objective: We aimed to describe the efficacy of CFTR modulators in the treatment of two siblings with the c.[1521_1523delCTT;1399C>T];[54-5940_273+10250del21kb] (L467F;F508del/CFTRdele2,3) genotype in clinical practice and *in vitro*. This article presents the clinical presentation and results of CFTR channel function assessment and personalized selection of CFTR modulators in monochorionic diamniotic twins with cystic fibrosis and the L467F;F508del/CFTRdele2,3 genotype. This is the first demonstration of the efficacy of a new CFTR modulator in patients with a complex allele and a class I variant in the genotype. The obtained results may be useful for choosing treatment strategies for patients with a complex allele and a class I variant in the genotype.

## 1. Introduction

Cystic fibrosis (CF) is an autosomal recessive disease caused by pathogenic variants in the cystic fibrosis transmembrane conductance regulator (*CFTR*) gene [1]. Multiorgan clinical manifestations of CFTR dysfunction in people with CF include structural damage and impaired lung function, as well as recurrent respiratory infections [2]. Progressive decline in lung function is a major cause of morbidity and mortality in people with CF. Clinical manifestations of lung pathology in CF include bronchitis and progressive airway obstruction, as well as bronchiectasis, with periods of increased respiratory symptoms against the background of exacerbations associated with an increased bacterial load in the respiratory tract. The clinical manifestations of CF and the burden of daily care also lead to a deterioration in the mental and physical health-related quality of life (HRQoL) of patients with cystic fibrosis, as well as their families and caregivers [3]. Over the past decade, advanced technologies have enabled the use of high-throughput screening approaches to drug discovery, yielding orally bioavailable small molecules capable of targeting the underlying defect—CFTR modulators [4,5].

CFTR modulators have significantly affected the prognosis for cystic fibrosis (CF), improving the clinical course in most patients with the F508del variant and several other *CFTR* gene variants. However, the variability in therapeutic response remains a relevant issue, which may be attributed to both comorbid factors and specific genotype features [6]. In particular, the presence of complex alleles, including that of more than one variant in the cis position, can alter protein properties and the efficacy of modulators [7,8,9].

An allele is considered complex if it carries at least two mutations in the *CFTR* gene on a single allele. In this case, each pathogenic variant can influence individual stages of CFTR biogenesis. Recently, it has become increasingly clear that complex alleles in the *CFTR* gene are not uncommon. Complex alleles can influence both the disease phenotype and susceptibility to targeted therapy, and assessing this influence remains challenging. The presence of a second variant on the same chromosome can either enhance [7] or weaken the effect of a pathogenic variant [10]. Furthermore, the presence of complex alleles complicates the ranking of CFTR mutations into different classes based on their functional effects. Knowledge of the presence of complex alleles in a patient is necessary for determining the prognosis of the disease, selecting a targeted therapy, and providing qualified medical genetic counseling to families.

One such case is the complex allele [L467F;F508del], in which the clinical response to therapy with the combination of elexacaftor/tezacaftor/ivacaftor (ETI) may be absent despite the presence of sensitivity in the case of an independent F508del variant [11]. Such alleles are often confirmed after the initiation of therapy with CFTR modulators.

Objective: We aimed to describe the efficacy of CFTR modulators in the treatment of two siblings with the genotype c.[1521_1523delCTT; 1399C>T];[54-5940_273+10250del21kb], p.[Phe508del;Leu467Phe];[Ser18Argfs*16] in clinical practice and *in vitro*.

## 2. Results

The patients were monochorionic diamniotic male twins born in 2017 from a normal pregnancy and an emergency operative delivery. At week 12 of gestation, ultrasound signs of an echogenic fetal bowel were observed in the fetuses, which were subsequently not confirmed. A positive neonatal screening result for both children was lost and obtained at the age of 5 months. No retest was performed due to missed due dates, and the parents declined a sweat test. The children had salty sweat, polyfecalia, and increased appetites since birth. There were no developmental delays; a cough appeared at the age of 4–5 months. The disorder was suspected at the age of 9 months based on the typical clinical signs, respiratory manifestations, intestinal syndrome and pseudo-Bartter syndrome that developed at this age, and confirmed by positive sweat tests (sweat conductivity values in the brothers were 127 mmol/L and 118 mmol/L) and molecular genetic testing (search for common CFTR gene mutations): the genotype F508del/CFTRdele2.3 (Del21kb) was determined.

Genotype c.[1521_1523delCTT; 1399C>T];[54-5940_273+10250del21kb], p.[Phe508del;Leu467Phe];[Ser18Argfs*16] has the traditional name [F508del;L467F]; [CFTR dele2,3], which is used hereinafter (Figure 1).

Following diagnosis, the children began receiving background therapy, including inhalation of dornase alfa, hypertonic NaCl solution, pancreatic enzymes, and kinesitherapy. Due to chronic respiratory tract infection with *S. aureus* (*MSSA*), intermittent cultures of *P. aeruginosa* were determined from the age of 2 years. Eradication courses of antibiotic therapy (oral ciprofloxacin and inhaled tobramycin) were prescribed, with a positive clinical and laboratory effect for two years. At the age of 4 years, a new culture of *P. aeruginosa* was determined. Frequent respiratory infections were noted from the age of 5. From the age of 6, frequent exacerbations of chronic bronchitis were reported, and oral antibacterial therapy was used. By the age of 6, nail clubbing was developed in both children.

At the age of 6, chest computed tomography (CT) showed traction bronchiectasis and linear pulmonary fibrosis in both children. The CT imaging of the paranasal sinuses revealed polypous rhinosinusitis. At the age of 6 years and 8 months, treatment with elexacaftor/tezacaftor/ivacaftor and ivacaftor (ETI) in the form of tablets in doses of 50 mg/25 mg/37.5 mg (2 tablets in the morning) and 75 mg (1 tablet in the evening) was initiated. No clear positive changes were reported during therapy. Respiratory episodes continued with the same frequency; the sweat test value increased after 6 months (Table 1). No adverse reactions were reported during therapy.

The intestinal current measurement (ICM) results obtained during ETI therapy for 3 weeks showed no restoration of chloride channel function (Table 2, Figure 2 and Figure 3). The change in the ΔI_SC_ in response to forskolin (to stimulate chloride channels) was 0 µA/cm^2^ in both patients, confirming the absence of CFTR channel function. ICM is a sensitive method for assessing the efficacy of treatment with CFTR modulators [12].

Given the lack of significant positive changes over time, additional genetic testing was performed. Using quantitative MLPA, we searched for the F508del and L467F nucleotide sequence variants, and analyzed the copy number of exon 11 of the *CFTR* gene (NM_000492) (Figure 4).

Analysis showed that both children have the pathogenic heterozygous F508del variant and the heterozygous L467F variant. There were two copies of exon 11 of the *CFTR* gene.

The presence of the complex allele [L467F; F508del] in a compound heterozygous state with the class VII CFTR dele2,3 (Del21kb) variant completely explains the failure of targeted ETI therapy [11,13,14].

A combination of vanzacaftor/tezacaftor/deutivacaftor may be considered as an alternative CFTR modulator. This is a new triple CFTR modulator administered once daily for the treatment of cystic fibrosis in patients aged 6 years and older who have at least one F508del mutation, another agent-sensitive mutation in the *CFTR* gene (https://www.cff.org/news/2024-12/fda-approves-new-cftr-modulator accessed on 22 October 2025) (FDA criteria) or at least one non-class I mutation in the *CFTR* gene (EMA criteria). The cumulative evidence from Phase 3 studies demonstrates that the next-generation CFTR modulator vanzacaftor + tezacaftor + deutivacaftor establishes a new standard of high-efficacy CFTR modulator therapy by providing superior CFTR ion channel function [15,16].

### Evaluation of the Effect of Targeted Agents on the Patient-Derived Intestinal Organoids

This study investigated the effect of the novel CFTR corrector vanzacaftor (VX-121) on the restoration of CFTR function in the [L467F;F508del]/CFTRdele2,3 genotype. It should be noted that the CFTRdele2,3 variant is a class VII pathogenic variant, and this study does not address its sensitivity to the tested targeted agents, as it disrupts CFTR mRNA formation. Two intestinal organoid cultures from twins were collected and subjected to a forskolin-induced swelling. A culture of organoids derived from a patient with the F508del/F508del genotype was used as a control. The independent effect of vanzacaftor was assessed, as was its effect in the triple combination of vanzacaftor/tezacaftor (VX-661)/ivacaftor (VX-770) (a commercial drug for the treatment of patients with CF that contains a deuterated derivative deutivacaftor instead of ivacaftor). In addition, the efficacy of vanzacaftor/tezacaftor/ivacaftor was compared with the dual combination of tezacaftor/ivacaftor, as well as with the triple combination of elexacaftor (VX-445)/tezacaftor/ivacaftor, which was the most effective, until recently, against the F508del/F508del genotype.

When stimulated with forskolin at a concentration of 5 µM, control organoids with the F508del/F508del genotype demonstrated a low level of swelling (AUC = 292 ± 47 relative units [RU]), which confirms the almost complete absence of CFTR function (Figure 5 and Figure 6). When exposed to VX-661 + VX-770 and VX-661 + VX-445 + VX-770, a pronounced increase in AUC values to 2085 ± 235 RU and 4519 ± 710 RU, respectively, was observed, whereas the organoids became enlarged by 60% (VX-661 + VX-770) and 120% (VX-661 + VX-445 + VX-770). Organoids derived from Patient 1 and Patient 2 demonstrated a complete lack of response to forskolin, indicating the absence of functional CFTR and consistent with the severity of the tested genotype (Figure 5 and Figure 6). A combination of VX-661 + VX-770 did not induce patient-derived organoid swelling (AUC values close to zero), demonstrating its *in vitro* inefficacy. Unlike VX-661 + VX-770, the triple combination of VX-661 + VX-445 + VX-770 (ETI) caused slight (20–30%) swelling of both organoid cultures with the [L467F;F508del]/CFTRdele2.3 genotype (AUC values were 718 ± 175 RU and 426 ± 102 RU for Patient 1 and Patient 2, respectively), proving the low sensitivity of the [L467F;F508del] complex allele to this combination. However, the magnitude of the response was significantly lower than that in the F508del/F508del control and did not reach a therapeutically significant level. The response to the new generation triple combination of VX-661 + VX-121 + VX-770 is of greatest interest. In both patients, stimulation with 5 µM forskolin in the presence of VX-661 + VX-121 + VX-770 resulted in significant organoid swelling (by 60–90%) and an AUC response of more than 2000 RU (2518 ± 290 for Patient 1; 2134 ± 514 for Patient 2), which exceeded the response to the triple-targeted therapy with VX-661 + VX-445 + VX-770 by approximately three times (Figure 5 and Figure 6).

## 3. Discussion

Loss-of-function *CFTR* gene mutations cause cystic fibrosis (CF) via various molecular mechanisms, including altered expression, trafficking, and/or activity of the CFTR chloride channel. The most frequent mutation among CF patients, F508del, causes multiple defects that can nevertheless be overcome by a combination of three pharmacological agents that improve CFTR channel trafficking and gating, namely elexacaftor, tezacaftor, and ivacaftor [13].

The new clinical case presented demonstrates the failure of therapy with elexacaftor/tezacaftor/ivacaftor and ivacaftor as a modulator in twin brothers with the [L467F;F508del]/CFTRdele2,3 genotype. The lack of clinical effect was confirmed by the persistence of exacerbations requiring antibacterial therapy and negative changes over time in pulmonary function in one of the brothers. The sweat test and the ICM showed no increase in the functional activity of the CFTR (chloride) channel. Moreover, the sweat test value was increased over time during ETI therapy in both brothers.

Previously, the lack of effect of tezacaftor/ivacaftor in the presence of a complex allele genotype, [L467F;F508del], was proven by us and other researchers [11,17,18].

The assumption that the patients have an additional *CFTR* gene mutation in the cis position was confirmed by MLPA. Both patients were found to have a complex allele [L467F;F508del], which interferes with the response to targeted therapy, as demonstrated by sweat testing and ICM. The frequency of the complex allele [L467F;F508del] in the Russian population, which is homozygous for the F508del variant, is 8.2% [14], and according to the 2023 registry data, the allele frequency among all CF patients is 0.87% (https://spulmo.ru/upload/WEB_Registre_2023.pdf accessed on 22 October 2025). The results obtained by ICM were confirmed in the intestinal organoid model. When stimulated with forskolin in the presence of ETI, patient-derived organoids with the [L467F;F508del]/CFTRdele2,3 genotype did not respond based on the lack of swelling required for the therapeutic effect of ETI to take place. A previous study conducted by Sondo et al. showed a similar result in two CF patients with a compound heterozygous F508del variant and a minimal-function class 1 variant (G542X and E585X) who did not experience any benefit after treatment with the ETI combination. Functional *in vitro* assays of the nasal epithelium from these patients confirmed the lack of response to treatment. Molecular diagnostics detected the L467F variant in the cis position with the F508del variant. The heterologous protein expression and Western blotting confirmed the lack of a positive effect on ETI to increase functional CFTR [13].

Previously, we presented a study of the complex allele [L467F;F508del] in a cohort with the [L467F;F508del] genotype/class I (c.3532_3535dup, c.1766+2T>C, W1310X, 712-1G>T), as well as in a patient with the [L467F;F508del]/[L467F;F508del] genotype. In intestinal organoids, the [L467F;F508del] homozygosity was shown to reduce the efficacy of dual treatment (ivacaftor + lumacaftor; ivacaftor + tezacaftor) and, in combination with class I variants, the efficacy of triple (ivacaftor + tezacaftor + elexacaftor) CFTR modulators.

In patients with the [L467F;F508del] genotype/class I, the results were confirmed by long-term (up to 6–12 months) treatment with ivacaftor + tezacaftor + elexacaftor in three patients without clinical effect [15].

This study is the next step in the further functional evaluation of complex alleles in terms of their effect on CFTR channel activity when using CFTR modulators. For the first time, the high efficacy of vanzacaftor/tezacaftor/ivacaftor (VNZ/TEZ/D-IVA) was demonstrated *in vitro* for patients with the [L467F;F508del]/CFTRdele2,3 genotype. It is important that for patients with the complex allele [L467F;F508del], which is common in the Russian Federation, a targeted drug has been developed that effectively restores the CFTR function of ion channels in these patients for the first time. The results will allow for further personalization of the use of CFTR modulators and optimization of their indications for use in the presence of the class I or VII variant in the genotype.

## 4. Materials and Methods

This paper presents in detail a familial case of CF in twins with the c.[1521_1523delCTT;1399C>T];[54-5940_273+10250del21kb], p.[Phe508del;Leu467Phe];[Ser18Argfs*16] genotype; the traditional name is [F508del;L467F]; [CFTRdele2,3]. These patients did not show a clinical response to ETI therapy.

The traditional nomenclature is used herein.

### 4.1. Molecular Genetic Testing

We extracted DNA from peripheral blood leukocytes using the QIAamp DNA Mini Kit, Qiagen, Hilden, Germany according to the manufacturer’s protocol. The F508del and CFTRdele2,3 mutations were identified by the amplification fragment length polymorphism using the search system for the most frequent 13 insertion/deletion mutations in the *CFTR* gene in the Russian population (c.54-5940_273+10250del21kb (CFTRdele2,3), c.1521_1523delCTT (F508del), c.2051_2052delAAinsG (2183AA>G), c.1545_1546delTA (1677delTA), c.2012delT (2143delT), c.2052dupA (2184insA), c.262_263delTT (394delTT), c.3691delT (3821delT), c.413_415dupTAC (L138ins), c.472_473insA (604insA), c.3816_3817delGT (3944delGT), c.3883delA (4015delA), c.3891dup (4022insT)) via multiplex amplification. The length of the amplified fragments ranged from 71 to 212 bp.

To identify the F508del and L467F mutations, as well as to analyze the copy number of exon 11 of the *CFTR* gene, quantitative multiplex ligation-dependent probe amplification (MLPA) was used, followed by amplification. TBP, B2M, and USP3 were used as control genes. The length of the obtained fragments ranged from 84 to 147 bp.

The oligonucleotide probes for ligation and primers for amplification were designed in the DNA Diagnostics Laboratory, Research Center for Medical Genetics.

The reference cDNA sequence posted on the NCBI website (http://www.ncbi.nlm.nih.gov/nuccore accessed on 22 October 2025), CFTR—NM_000492.4, was used.

### 4.2. Intestinal Current Measurement

The study of the intestinal current measurement (ICM) method was carried out according to the European standard operating procedures V2.7_26.10.11 (SOPs) [19,20].

### 4.3. Forskolin-Induced Swelling in Intestinal Organoids

Intestinal crypts were isolated from human biopsy samples. For this purpose, the crypts were incubated with a solution of 10 mM EDTA (Thermo Fisher Scientific: Invitrogen, Waltham, MA, USA), and then immersed in a Matrigel matrix (Corning, Bedford, MA, USA) and cultured in 24-well plates (SPL Life Sciences, Yeoju-si, Republic of Korea). After the polymerization of the Matrigel, a growth medium was added. The composition of the medium is described in the [21]. The organoids were subcultured once every 7 days by mechanical dissociation of large budding structures into small fragments. To perform forskolin-induced swelling, organoids were cultured in 96-well plates. After 24 h, the organoids were stained with Calcein AM (0.85 µM; BioLegend, San Diego, CA, USA) and stimulated with forskolin at a concentration of 5 µM. The treatment continued for 60 min. At certain time points (every 10 min for one hour), fixed fields were recorded using a Axio Observer 7 Fluorescence microscope (Zeiss, Oberkochen, Germany). The correctors tezacaftor VX-661, elexacaftor VX-445, and vanzacaftor VX-121 (3.5 µM; Selleckchem, Houston, TX, USA) were added to the growth medium at the stage of organoid culturing, and the potentiator ivacaftor VX-770 (3.5 µM; Selleckchem, Houston, TX, USA) was added concomitantly with forskolin. Quantitative analysis of organoid swelling was performed using custom-written Analyzer software. Results are presented as the area under the curve (AUC) in absolute units and expressed as the mean ± standard deviation.

## 5. Conclusions

This case demonstrates a personalized approach to CFTR modulator therapy for patients with CF. This approach includes comprehensive molecular genetic testing for the F508del variant, using sequencing to search for complex alleles, as well as the ICM and FIS (forskolin-induced swelling) methods, on the patient-derived intestinal organoids. A similar approach is applied to carriers of other genetic variants who have failed targeted therapy.

Using the intestinal organoid model and ICM method, the complex allele [L467F;F508del] in a heterozygous state with CFTRdele2,3, carried by the twin brothers, was shown to be characterized by the absence of CFTR channel function and the absence of a clinically significant effect of ETI. Significantly improved functional restoration *in vitro* of the novel drug VNZ/TEZ/IVA for this genotype was demonstrated using intestinal organoids for the first time, which indicates that there is promise in using this combination of CFTR modulators for patients with the [L467F;F508del] genotype/classes I or VII. However, confirmation of improved CFTR protein maturation by Western blot analysis is needed in the future.

## Figures and Tables

**Figure 1 ijms-26-11742-f001:**
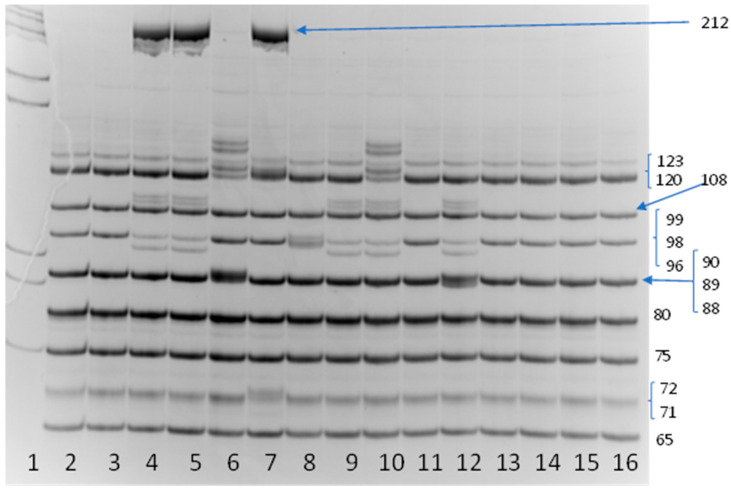
Electropherogram of the amplification fragment length polymorphism (AFLP) for recording 13 common CFTR gene mutations. Lane 1: molecular weight marker λ/Pst1. Lanes 2, 3, 11, 13, 14, 15, 16: normal. Lane 4 and Lane 5: heterozygous F508del mutation and heterozygous CFTRdele2,3 (Del21kb) mutation. Lane 6: heterozygous c.413_415dupTAC (L138ins) mutation and heterozygous c.2052dupA (2184insA) mutation. Lane 7: heterozygous c.472_473insA (604insA) mutation and heterozygous CFTRdele2,3 (Del21kb) mutation. Lane 8: heterozygous c.1515del (p.Asn505Lysfs*22) mutation. Lane 9: heterozygous F508del mutation. Lane 10: heterozygous c.413_415dupTAC (L138ins) mutation and heterozygous F508del mutation. Lane 12: heterozygous F508del mutation and heterozygous c.2012delT (2143delT) mutation.

**Figure 2 ijms-26-11742-f002:**
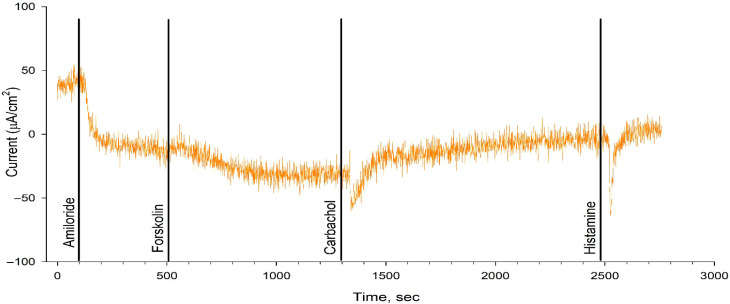
ICM results for Patient 1 following 3-week treatment with ETI.

**Figure 3 ijms-26-11742-f003:**
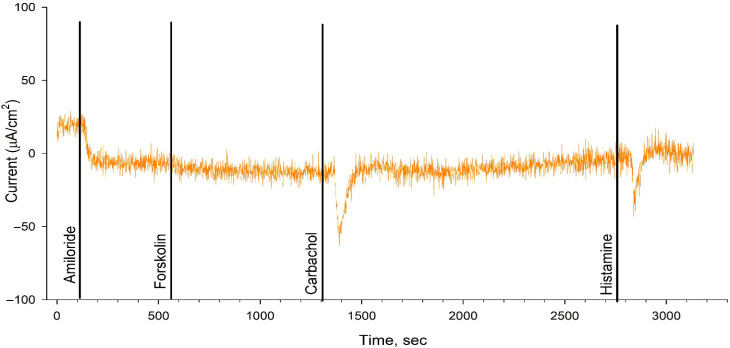
ICM results for Patient 2 following 3-week treatment with ETI.

**Figure 4 ijms-26-11742-f004:**
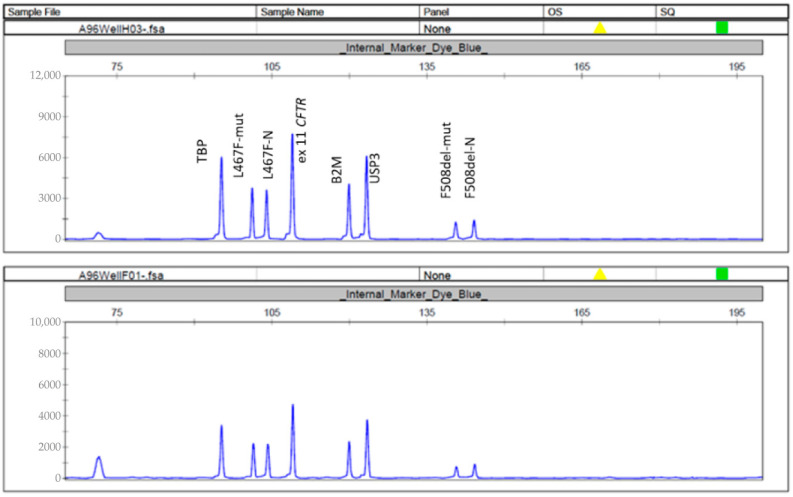
Fragment analysis. This figure presents the system we used for the recording of L467F and F508del and the determination of the number of copies of exon 11 of the *CFTR* gene by MLPA. The heterozygous variants of L467F and F508del were identified in the siblings.

**Figure 5 ijms-26-11742-f005:**
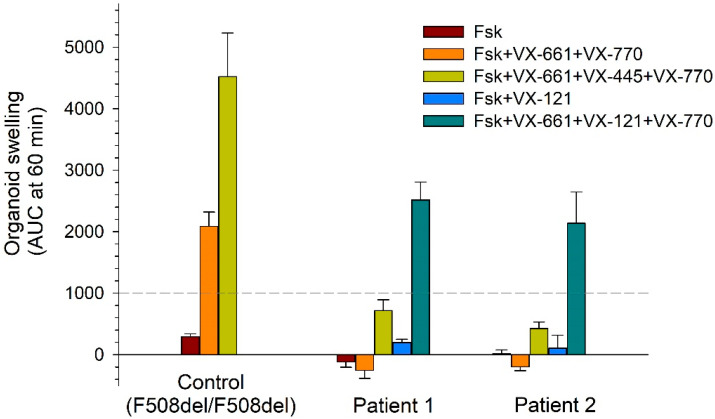
Assessment of the forskolin-induced swelling in studying the effect of CFTR modulators on the restoration of [L467F;F508del]–CFTR protein function. Control: organoid culture with the F508del/F508del genotype. Concentrations: 3.5 µM for VX-661, VX-121, VX-445 and VX-770; 5 µM for Fsk.

**Figure 6 ijms-26-11742-f006:**
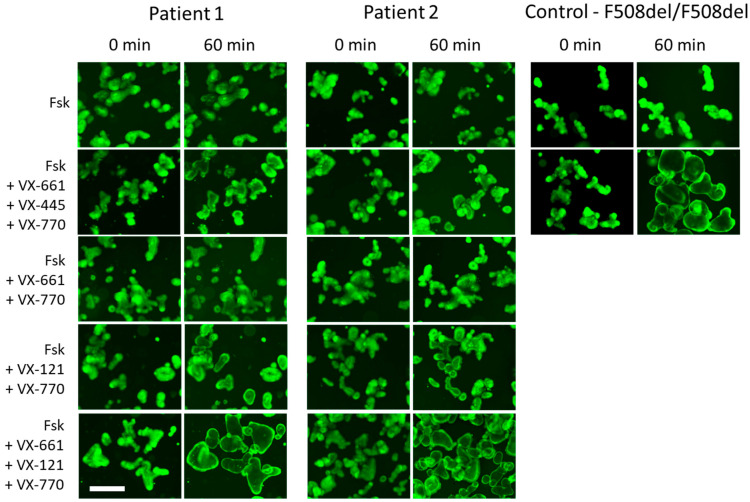
Representative micrographs obtained during the assessment of forskolin-induced swelling of intestinal organoids with the [L467F;F508del]/CFTRdele2.3 genotype in comparison with F508del/F508del organoids. Organoids were stained with calcein, lens: ×5, scale bar: 200 µm.

**Table 1 ijms-26-11742-t001:** Clinical and laboratory findings of patients before and during therapy with the ETI CFTR modulator.

Parameter	Patient 1	Patient 2	Comments
Immunoreactive trypsin (4 days of life)	105 ng/mL	135 ng/mL	Cut-off for IRT—65 ng/mL
First symptoms	Echogenic fetal bowel at week 12 of gestation; increased appetite, polyfecalia salty sweat since birth	Echogenic fetal bowel at week 12 of gestation; increased appetite, polyfecalia, salty sweat since birth	At the age of 5 months, a cough appeared during teething
Age at diagnosis	9 months	9 months	Pseudo-Bartter syndrome
Sweat test, conductivity (mmol/L NaCl)	101; 127	119; 98	Positive: >80 mmol/L
Pancreatic elastase, µg/g	<15	<15	Severe pancreatic insufficiency (normal: over 200 µg/g stool)
Initial genotype (9 months)	F508del/CFTRdele2.3		Search for common mutations
Comorbidities	Benign bilirubinemia, Gilbert’s syndrome?	Background retinopathy. Congenital heart defect: Bicuspid aortic valve. Aortic valve insufficiency, stage 1.5–2. Mild aortic stenosis. Cardiac conduction disorder: sinoatrial block, stage 2; accelerated QT conduction	–
Changes in parameters over time during ETI therapy (before ETI → in 9 months)			
Weight, kg	20.4→24	20.5→22.7	–
Height, cm	125→127	124.5→126	–
BMI, kg/m^2^	13.12→14	13.3→13.6	–
Sweat test, mmol/L	127→135	124→138	Negative changes
FEV1 (% of normal)	100→110	118→89	–
Microbiological status	*E. coli*→*MSSA*	*MSSA + S. maltophilia*→ *MSSA*	–
ALT/AST, U/L	24.2→23.6 34.7→30.3	22.6→20 48→35	–
Total bilirubin, µmol/L	25.3→24.9	34.4→22.5	–
Number of exacerbations per year requiring antibacterial therapy	3→3 (milder course)	2–3→2 (with prolonged rhinitis)	–
General condition	Improved fatigue, weight gain	No changes	–
Adverse reactions	None	None	–

**Table 2 ijms-26-11742-t002:** Short-circuit current density (ΔI_SC_) values during stimulant administration in each patient.

ΔI_SC_, M ± m µA/cm^2^	Amiloride	Forskolin	Carbahol	Histamine
Patient 1	−25.67 ± 7.84	0	32.17 ± 6.58	23.67 ± 5.48
Patient 2	−16 ± 7.84	0	22 ± 6.6	20.17 ± 3.54
F508del/F508del [12]	−18.39 ± 5.62	3.06 ± 0.89	16.55 ± 1.44	21.5 ± 5.46
Healthy control (wt/wt CFTR) [12]	−8.98 ± 3.42	25.78 ± 4.41	117.44 ± 4.32	101.68 ± 10.99

## Data Availability

The datasets used and/or analyzed during the current study are available from the corresponding author upon reasonable request.

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
