# Peer review of "L467F;F508del Complex Allele in a Heterozygous State with CFTRdele2,3: What to Expect from CFTR Modulators?"

_ijms, 2025, doi:10.3390/ijms262311742_

Round 1
Reviewer 1 Report
Comments and Suggestions for Authors
This manuscript presents valuable data demonstrating the effects of CFTR modulators on rare CFTR mutations, providing important insights into the field of cystic fibrosis. However, there are several major issues that must be addressed before publication.
Introduction:
The introduction is too brief and lacks sufficient background information about cystic fibrosis and CFTR modulators. A substantial revision and expansion of this section are required, as the current version has serious shortcomings.
Figure 1 (AFLP analysis):
A gene map indicating the restriction enzyme cutting sites should be included, along with a clear illustration of the expected fragment sizes.
Figure 3:
The manuscript lacks a citation or reference to Figure 3. This should be added.
CFTRdele2,3 mutation:
Additional information should be provided regarding the extent of amino acid deletion caused by the CFTRdele2,3 mutation.
Author Response
Thank you very much for such a thorough analysis of our manuscript.
Comments: This manuscript presents valuable data demonstrating the effects of CFTR modulators on rare CFTR mutations, providing important insights into the field of cystic fibrosis. However, there are several major issues that must be addressed before publication.
Introduction:
The introduction is too brief and lacks sufficient background information about cystic fibrosis and CFTR modulators. A substantial revision and expansion of this section are required, as the current version has serious shortcomings.
Response: Thank you very much for such a thorough analysis of our manuscript. We have added information about the disease and modulators.
Comments: Figure 1 (AFLP analysis):
A gene map indicating the restriction enzyme cutting sites should be included, along with a clear illustration of the expected fragment sizes.
Response: Thank you for your comment! No restriction enzymes were used in this study. To detect the F508del and CFTRdele2,3 (Del21kb) mutations, a method called amplification fragment length polymorphism (AFLP) was used (primers flank the mutations under study, and the AFLPs are separated on a polyacrylamide gel based on their length differences).
[see file for image]
Electropherogram of the amplification fragment length polymorphism (AFLP) for recording 13 common CFTR gene mutations.
Lane 1: molecular weight marker λ/Pst1.
Lanes 2, 3, 11, 13, 14, 15, 16: normal.
Lane 4 and Lane 5: heterozygous F508del mutation and heterozygous CFTRdele2,3 (Del21kb) mutation.
Lane 6: heterozygous c.413_415dupTAC (L138ins) mutation and heterozygous c.2052dupA (2184insA) mutation.
Lane 7: heterozygous c.472_473insA (604insA) mutation and heterozygous CFTRdele2,3 (Del21kb) mutation.
Lane 8: heterozygous c.1515del (p.Asn505Lysfs*22) mutation.
Lane 9: heterozygous F508del mutation.
Lane 10: heterozygous c.413_415dupTAC (L138ins) mutation and heterozygous F508del mutation.
Lane 12: heterozygous F508del mutation and heterozygous c.2012delT (2143delT).
Comments: Figure 3:
The manuscript lacks a citation or reference to Figure 3. This should be added.
Response: Thank you. Corrected it.
Comments: CFTRdele2,3 mutation:
Additional information should be provided regarding the extent of amino acid deletion caused by the CFTRdele2,3 mutation.
Response: The variant of the nucleotide sequence c.54-5940_273+10250del21kb results in a shift in the reading frame with the formation of a premature termination codon p.(Ser18Argfs*16).

Reviewer 2 Report
Comments and Suggestions for Authors
This manuscript presents a case study of twin brothers with the complex [L467F;F508del]/CFTRdele2,3 genotype. The study is divided into two parts: (1) a confirmation of the clinical and functional failure of ETI (elexacaftor/tezacaftor/ivacaftor) therapy, and (2) a new, in vitro proposal of an alternative therapy with a triple combination based on the new corrector vanzacaftor.
While the clinical data could be valuable, the manuscript’s novel claims about vanzacaftor suffer from a significant lack of supporting experimental evidence, particularly when viewed in the context of the mechanistic work already published by Sondo et al. (2022).
This manuscript confirms the same ETI failure in two new patients. It supports this with strong in vivo clinical data (no improvement in symptoms, negative changes in sweat tests) and functional data (no CFTR function in Intestinal Current Measurements). The in vitro organoid assay also showed ETI's inefficacy. In short, the Kondratyeva paper validates the Sondo et al. findings in a new patient cohort and with a different (but complementary) set of functional assays.
The primary novel contribution of the Kondratyeva paper is the claim that a new Vanzacaftor-based combination (VNZ/TEZ/IVA) shows "exceptionally high efficacy" for this genotype. This central conclusion is premature and based on a significant lack of experimental evidence.
The entire basis for this claim is one experiment: a forskolin-induced swelling (FIS) assay on patient-derived intestinal organoids. While the organoids showed significant swelling with the vanzacaftor combination, this is a functional "black box" assay. The Kondratyeva manuscript is missing the crucial biochemical experiments (e.g., Western blots) to demonstrate that the Vanzacaftor combination (unlike ETI) successfully rescues the maturation and trafficking of the L467F-F508del-CFTR protein. Without this, the "efficacy" is just an observation from one functional assay, lacking a demonstrated biological mechanism.
In addition, the patients in this study were never administered the vanzacaftor combination. The paper demonstrates a clinical failure of ETI but only hypothesizes the success of vanzacaftor based on an in vitro test. To support the claim of "exceptionally high efficacy", the manuscript would need to provide the necessary mechanistic evidence (e.g., Western blots) showing that Vanzacaftor, unlike ETI, can overcome the severe maturation defect of the L467F-F508del protein that was previously identified by Sondo et al. (2022) . Without this, the findings remain preliminary.
Author Response
Comments: This manuscript presents a case study of twin brothers with the complex [L467F;F508del]/CFTRdele2,3 genotype. The study is divided into two parts: (1) a confirmation of the clinical and functional failure of ETI (elexacaftor/tezacaftor/ivacaftor) therapy, and (2) a new, in vitro proposal of an alternative therapy with a triple combination based on the new corrector vanzacaftor.
While the clinical data could be valuable, the manuscript’s novel claims about vanzacaftor suffer from a significant lack of supporting experimental evidence, particularly when viewed in the context of the mechanistic work already published by Sondo et al. (2022).
This manuscript confirms the same ETI failure in two new patients. It supports this with strong in vivo clinical data (no improvement in symptoms, negative changes in sweat tests) and functional data (no CFTR function in Intestinal Current Measurements). The in vitro organoid assay also showed ETI's inefficacy. In short, the Kondratyeva paper validates the Sondo et al. findings in a new patient cohort and with a different (but complementary) set of functional assays.
The primary novel contribution of the Kondratyeva paper is the claim that a new Vanzacaftor-based combination (VNZ/TEZ/IVA) shows "exceptionally high efficacy" for this genotype. This central conclusion is premature and based on a significant lack of experimental evidence.
The entire basis for this claim is one experiment: a forskolin-induced swelling (FIS) assay on patient-derived intestinal organoids. While the organoids showed significant swelling with the vanzacaftor combination, this is a functional "black box" assay. The Kondratyeva manuscript is missing the crucial biochemical experiments (e.g., Western blots) to demonstrate that the Vanzacaftor combination (unlike ETI) successfully rescues the maturation and trafficking of the L467F-F508del-CFTR protein. Without this, the "efficacy" is just an observation from one functional assay, lacking a demonstrated biological mechanism.
In addition, the patients in this study were never administered the vanzacaftor combination. The paper demonstrates a clinical failure of ETI but only hypothesizes the success of vanzacaftor based on an in vitro test. To support the claim of "exceptionally high efficacy", the manuscript would need to provide the necessary mechanistic evidence (e.g., Western blots) showing that Vanzacaftor, unlike ETI, can overcome the severe maturation defect of the L467F-F508del protein that was previously identified by Sondo et al. (2022). Without this, the findings remain preliminary.
Response: Thank you very much for such a thorough analysis of our manuscript. The data obtained in this study allow future use of targeted therapy with vanzacaftor/tezacaftor/deutivacaftor for patients with the L467F;F508del complex allele and the class 1 genotype variant. This drug is not registered in our country, and we cannot prescribe it at this time. Western blots may be performed after 3 months due to the length of organoid culture and the waiting list for the Western blots itself to evaluate the CFTR protein. We planned to perform Western blots in our future work with a larger number of patients.
Round 2
Reviewer 1 Report
Comments and Suggestions for Authors
The authors appear to have addressed the previously raised issues to a reasonable extent. Although the introduction still feels somewhat insufficient, the manuscript is acceptable for publication.
Author Response
Comments: The authors appear to have addressed the previously raised issues to a reasonable extent. Although the introduction still feels somewhat insufficient, the manuscript is acceptable for publication.
Response: Thank you very much for appreciating the work. We have also added information about the disease and modulators. We attach the article with corrections.

Reviewer 2 Report
Comments and Suggestions for Authors
The main concern raised in the initial review remains: the claim of "exceptionally high efficacy" for the VNZ-based combination is based entirely on a single functional assay (FIS) and lacks the crucial mechanistic evidence to explain why this new modulator combination succeeds where ETI fails.
Missing Mechanistic Data: The previous work by Sondo et al. highlighted that the variant, when in cis with , causes a severe maturation/trafficking defect that ETI cannot overcome. To conclusively support the superiority of the VNZ/TEZ/IVA combination, the manuscript still needs to provide biochemical evidence (e.g., Western blotting) demonstrating that the VNZ-based combination successfully rescues the maturation and trafficking of the -CFTR protein to the cell surface, producing the mature Band C form.
Response to Reviewer: The authors’ response mentions that Western blots are planned for future work due to logistical constraints. While understandable, this absence means the study remains a strong observational functional finding (in vitro) but lacks a mechanistic explanation needed to fully support the "exceptionally high efficacy" claim.
While awaiting the mechanistic data, the authors should temper the language regarding the VNZ-based combination in the Abstract, Introduction, and Conclusion. Phrases like "exceptionally high efficacy" should be modified to "significantly improved functional restoration in vitro" or similar. They should explicitly state the need for future Western blot analysis to confirm the molecular mechanism (i.e., improved protein maturation/trafficking).
Author Response
Comments: While awaiting the mechanistic data, the authors should temper the language regarding the VNZ-based combination in the Abstract, Introduction, and Conclusion. Phrases like "exceptionally high efficacy" should be modified to "significantly improved functional restoration in vitro" or similar. They should explicitly state the need for future Western blot analysis to confirm the molecular mechanism (i.e., improved protein maturation/trafficking).
Response: Thank you very much for your comments. We have made corrections to the conclusion. We attach the article with corrections.

Round 3
Reviewer 2 Report
Comments and Suggestions for Authors
The authors have addressed my concerns